# Long Non-Coding RNA *ANRIL* as a Potential Biomarker of Chemosensitivity and Clinical Outcomes in Osteosarcoma

**DOI:** 10.3390/ijms222011168

**Published:** 2021-10-16

**Authors:** Adam M. Lee, Asmaa Ferdjallah, Elise Moore, Daniel C. Kim, Aritro Nath, Emily Greengard, R. Stephanie Huang

**Affiliations:** 1Department of Experimental and Clinical Pharmacology, College of Pharmacy, University of Minnesota, Minneapolis, MN 55455, USA; leeam@umn.edu (A.M.L.); kim00278@umn.edu (D.C.K.); 2Department of Pediatrics, Hematology & Oncology, University of Minnesota, Minneapolis, MN 55455, USA; asmaa@umn.edu (A.F.); emilyg@umn.edu (E.G.); 3Department of Natural Sciences, Zanvyl Krieger School of Arts and Sciences, The Johns Hopkins University, Baltimore, MD 21218, USA; emoore40@jhu.edu; 4Department of Medical Oncology and Therapeutics, City of Hope Comprehensive Cancer Center, Monrovia, CA 91007, USA; anath@coh.org

**Keywords:** osteosarcoma, cisplatin, doxorubicin, long non-coding RNA, *ANRIL*, CDKN2B-AS1, chemosensitivity

## Abstract

**Simple Summary:**

Osteosarcoma is a bone cancer with a poor prognosis. This is, in part, due to resistance to current standard-of-care chemotherapeutic treatment. As personalized treatment plans become more widely utilized, the role of patient-specific genome markers may serve to identify individuals with chemo-resistant disease at the outset of diagnosis. *ANRIL*, a long non-coding RNA, has promise as a predictive biomarker. Utilizing osteosarcoma cell lines, we observed that altering the expression of *ANRIL* significantly alters the sensitivity to cisplatin and doxorubicin, two agents that are a standard-of-care for treatment. Analysis of clinical data from the TARGET dataset confirmed higher *ANRIL* expression portending poorer prognosis, as evidenced by association with death and metastases at diagnosis.

**Abstract:**

Osteosarcoma has a poor prognosis due to chemo-resistance and/or metastases. Increasing evidence shows that long non-coding RNAs (lncRNAs) can play an important role in drug sensitivity and cancer metastasis. Using osteosarcoma cell lines, we identified a positive correlation between the expression of a lncRNA and *ANRIL,* and resistance to two of the three standard-of-care agents for treating osteosarcoma—cisplatin and doxorubicin. To confirm the potential role of *ANRIL* in chemosensitivity, we independently inhibited and over-expressed *ANRIL* in osteosarcoma cell lines followed by treatment with either cisplatin or doxorubicin. Knocking-down *ANRIL* in SAOS2 resulted in a significant increase in cellular sensitivity to both cisplatin and doxorubicin, while the over-expression of *ANRIL* in both HOS and U2OS cells led to an increased resistance to both agents. To investigate the clinical significance of *ANRIL* in osteosarcoma, we assessed *ANRIL* expression in relation to clinical phenotypes using the osteosarcoma data from the Therapeutically Applicable Research to Generate Effective Treatments (TARGET) dataset. Higher *ANRIL* expression was significantly associated with increased rates of metastases at diagnosis and death and was a significant predictor of reduced overall survival rate. Collectively, our results suggest that the lncRNA *ANRIL* can be a chemosensitivity and prognosis biomarker in osteosarcoma. Furthermore, reducing *ANRIL* expression may be a therapeutic strategy to overcome current standard-of-care treatment resistance.

## 1. Introduction

Osteosarcoma is a skeletal neoplasm derived from primitive mesenchymal cells that affects children, adolescents, and young adults [1]. Although only representing 5% of pediatric malignancies, it is the most common primary solid malignancy of bone and has the highest incidence in adolescence [1,2]. A multimodal approach consisting of neoadjuvant chemotherapy and surgical resection is the mainstay of therapy [3,4]. Patients receive an initial two cycles of chemotherapy with the agents cisplatin, doxorubicin, and methotrexate [3]. Following this, patients undergo resection of the primary tumor, which is then assessed for extent of necrosis [4]. Patients then complete therapy with four more cycles of cisplatin, doxorubicin, and methotrexate [3].

Prior to the introduction, widespread use, and standardization of these chemotherapy agents, 5-year overall survival for osteosarcoma was poor at 20% [5]. Greater than 90% of patients died from pulmonary metastases [6]. The addition of neoadjuvant chemotherapy to surgical resection (or amputation) increased survival rates to 60–70% [7]. This initial increase in 5-year overall survival suggests that chemosensitivity, at least in part, can perhaps be viewed as a prognostic factor. However, this means that despite chemotherapy and surgical resection, 30–40% of patients do not survive beyond 5 years [8]. This is primarily due to the lack of initial response to the standard chemotherapy regimen of cisplatin, doxorubicin, and methotrexate, and demonstrates the urgent need to identify biomarkers that can be used to predict chemosensitivity [9].

Current known prognostic biomarkers for osteosarcoma, including *Rb* gene and/or *TP53* mutations, are shown to be related to the incidence and outcomes of disease but are poor markers to predict chemosensitivity [10]. Therefore, the percent necrosis of tumor tissue after two cycles of chemotherapy has been used as a clinical biomarker of chemosensitivity [4,11,12]. However, a major drawback of this approach is that it requires patients to undergo toxic and potentially ineffective chemotherapy before determining resistance. Furthermore, this chemotherapy-only response predictor is no longer deemed sufficient to determine chemoresistance by many clinicians despite its initial promising use in The Children’s Oncology Group osteosarcoma trial [13].

Non-coding genes are a growing area of interest and many non-coding RNAs have been found to act as oncogenes or tumor suppressors [14]. LncRNAs (long non-coding RNA) make up 60% of the human genome and are characterized by their length of greater than 200 nucleotides [15]. They do not encode proteins but can manipulate local or global gene expression via transcription in addition to post-transcriptional and epigenetic regulation [16]. These manipulations can lead to either the inhibition or activation of protein-coding genes. To date, less than 1% of lncRNAs have been functionally characterized [17]. Given this, along with the fact that there are limited protein-coding gene mutations observed in pediatric cancers, lncRNAs present an exciting opportunity for clinical biomarker discovery as epigenetic regulators of cancer. Previous studies indicated that a growing number of lncRNAs play a key role in osteogenic differentiaton [18] and they were found to be associated with bone tumor onset and development [19,20]. Taken together, lncRNAs may serve as novel biomarkers for chemotherapeutic sensitivity in osteosarcoma.

The current knowledge of lncRNAs in terms of the initiation and development of human cancers has grown significantly, whereas investigations into the role of lncRNAs in regulating the sensitivity of cancer cells to antineoplastic treatments have just begun to make significant progress [21,22,23]. Recently, our research group performed a systematic analysis of the association between the somatic lncRNA transcriptome and genome of hundreds of cell lines with pharmacological profiles for hundreds of drugs, utilizing the Genomics of Drug Sensitivity in Cancer database (GDSC) [24,25,26]. LncRNAs were found to be just as potent as the protein-coding transcriptome at drug response prediction. In some instances, lncRNA outperformed known protein-coding gene biomarkers in predicting drug response. Among them, we observed significant associations between the lncRNA *ANRIL* (anti-sense non-coding RNA in the INK4 locus) and sensitivity to multiple drugs, including the mainstays of osteosarcoma treatment, cisplatin and doxorubicin (Appendix A Appendix A).

The lncRNA *ANRIL* (also known as *CDKN2B-AS1*) was first identified from familial melanoma patients with a large germline deletion in the INK4B-ARF-INK4A gene cluster, and has been previously characterized as an oncogene in osteosarcoma and other cancer types [20,27,28,29,30]. In relation to chemotherapeutic response, previous studies have shown *ANRIL* to promote cell chemoresistance in multiple cancers; however, the impact of *ANRIL* in response to cisplatin and doxorubicin in the treatment of osteosarcoma is not fully elucidated [20,31,32,33,34,35,36,37].

Based on our previous analysis and the current existing evidence, we hypothesized that *ANRIL* may play a role in the variability of therapeutic response to cisplatin and doxorubicin and could serve as a potential clinical biomarker of therapeutic response in osteosarcoma. To investigate the relationship between *ANRIL* and chemotherapeutic sensitivity, this study utilized osteosarcoma cell lines to develop *ANRIL* knockdown and overexpression models to determine the effects on cell viability during cisplatin and doxorubicin treatment. To investigate the potential clinical utility of *ANRIL*, we also utilized expression data and clinical outcome measures in a publicly available osteosarcoma patient cohort.

## 2. Results

### 2.1. Higher ANRIL Expression Level Is Correlated with Cisplatin and Doxorubicin Resistance in Osteosarcoma Cell Lines

Using data from a publicly available high-throughput cancer cell line drug screening dataset, GDSC1 [38], we found significant correlations between the IC50 values of cisplatin and doxorubicin and *ANRIL* expression (Appendix A) in a collection of cancer cell lines. When focusing these analyses only on available osteosarcoma cell lines from this GDSC1 screen, we again found positive correlations between the expression level of *ANRIL* and both drugs’ IC50 values (Figure 1). The correlation between *ANRIL* expression and IC50 was statistically significant for cisplatin but did not reach statistical significance for doxorubicin due to the small sample size.

### 2.2. ANRIL Knockdown in SAOS2 and Impact on Cell Viability during Cisplatin and Doxorubicin Treatment

Knockdown of the lncRNA *ANRIL* in SAOS2 cells using siRNA displayed an 88.9% decrease in *ANRIL* expression compared to the SAOS2 cells transfected with the control siRNA at 24 h following transfection (Figure 2A).

The siRNA knockdown effect on *ANRIL* expression was maintained up to 72 h post-transfection at >80% knockdown efficiency. In order to assess the specificity of siRNA *ANRIL* knockdown, we additionally measured the expression of three neighboring protein-coding genes, *CDKN2A*, *CDKN2B*, and *MTAP*, and found no significant expression differences between SAOS2 cells transfected with *ANRIL* siRNA and SAOS2 cells transfected with control siRNA (Figure 2B,C). To determine the cellular response to decreased *ANRIL* expression, we measured cell viability at 24, 48, and 72 h post-transfection in SAOS2 cells transfected with either *ANRIL* or control siRNA using the WST-1 reagent. We observed a lower rate of cell proliferation (calculated by comparing cell viability at various post-treatment time points to those obtained at baseline) for SAOS2 *ANRIL* knockdown cells when compared to the control, but the difference was not statistically significant (Appendix A).

To assess the impact of decreased *ANRIL* expression on cisplatin and doxorubicin sensitivity, we treated SAOS2 cells transfected with either *ANRIL* or control siRNA with serial dilutions of cisplatin and doxorubicin, and measured cell viability at 24, 48, and 72 h post-treatment using the cell viability reagent WST-1. We observed a significant difference in cell viability between *ANRIL* knockdown SAOS2 cells compared to the control assessed at 72 h post-treatment with either cisplatin or doxorubicin (two-way ANOVA, *p* < 0.0001 for both treatments). The siRNA *ANRIL* knockdown SAOS2 cells displayed a significantly decreased cisplatin IC50 of 3.6 uM compared to the siRNA control SAOS2 cells, which had a cisplatin IC50 of 5.3 uM (Figure 3A).

siRNA *ANRIL* knockdown SAOS2 cells displayed a significantly decreased doxorubicin IC50 of 165 nM compared to siRNA control SAOS2 cells with a doxorubicin IC50 of 302.7 nM (Figure 3B). Comparisons of cell viability in SAOS2 cells transfected with either *ANRIL* or control siRNA, at 24 and 48 h post-treatment with either cisplatin or doxorubicin, are provided in Appendix A. Significant differences in drug sensitivity between ANRIL knockdown and control models were observed at both 24 h and 48 h post-treatment with cisplatin (Appendix A) and doxorubicin (Appendix A).

### 2.3. ANRIL Overexpression in HOS and U2OS and Impact on Cell Viability during Cisplatin and Doxorubicin Treatment

Stable *ANRIL* overexpression models using HOS and U2OS cell lines showed significant increases in *ANRIL* expression compared to the HOS and U2OS expression vector controls with a 114-fold increase observed in HOS and a 60-fold increase in U2OS (Figure 4A,D).

The assessment of *CDKN2A*, *CDKN2B*, and *MTAP* expression showed no significant differences between *ANRIL* overexpression HOS and U2OS models compared to their respective controls (Figure 4B,C,E,F). To determine if increased *ANRIL* expression promotes cell growth, we measured cell viability at 24, 48, and 72 h post-transfection in *ANRIL* overexpression and vector control models in both the HOS and U2OS cell lines using the cell viability reagent WST-1. We observed an increased rate of proliferation over time in both the HOS and U2OS *ANRIL* overexpression models compared to their respective controls in the absence of cisplatin and doxorubicin treatment (Figure 5A,B).

To assess the impact of increased *ANRIL* expression on cisplatin and doxorubicin sensitivity, we treated stable *ANRIL* overexpression and expression control models for HOS and U2OS cells with serial dilutions of cisplatin and doxorubicin, and measured cell viability at 24, 48, and 72 h post-treatment using the cell viability reagent WST-1. We observed a significant difference in cell viability between *ANRIL* overexpression U2OS cells compared to the control assessed at 72 h post-treatment with either cisplatin or doxorubicin (two-way ANOVA, *p* < 0.0001 for both treatments). Stable *ANRIL* overexpression in U2OS cells displayed a significantly increased cisplatin IC50 of 12.7 uM compared to the expression vector control U2OS cells with a cisplatin IC50 of 9.0 uM (Figure 6A).

Stable *ANRIL* overexpression in U2OS cells displayed a significantly increased doxorubicin IC50 of 239.4 nM compared to expression vector control U2OS cells with a doxorubicin IC50 of 146.2 nM (Figure 6B). In HOS cells, although we observed a similar trend of decreased sensitivity to cisplatin and doxorubicin at 72 h post-treatment in *ANRIL* overexpression HOS cells compared to the control (Figure 6C,D), the difference was not statistically significant. Comparisons of cell viability in both U2OS and HOS overexpression models at 24 and 48 h post-treatment with either cisplatin or doxorubicin are provided in Appendix A. Significant differences in drug sensitivity between ANRIL overexpression U2OS cells and vector control were observed at both 24 h and 48 h post-treatment with cisplatin (Appendix A) and doxorubicin (Appendix A). HOS cells containing the ANRIL overexpression vector displayed no significant decreases in sensitivity to cisplatin at either 24 or 48 h post-treatment (Appendix A), but did display a significant decrease in sensitivity to doxorubicin at 24 h post-treatment (Appendix A). The observed change in doxorubicin sensitivity between ANRIL overexpression HOS cells and control were not maintained in the 48 h post-treatment measurement (Appendix A).

### 2.4. ANRIL Expression in Osteosarcoma Patients and Association with Clinical Outcome Measures

The role of lncRNA expression in osteosarcoma prognosis was examined utilizing the TARGET osteosarcoma database. Survival outcomes, metastases at diagnosis, and percent necrosis were evaluated in osteosarcoma patients (*n* = 86 and *n* = 43 for OS/metastases and percent necrosis, respectively) with various *ANRIL* expression levels. We found that higher *ANRIL* expression was significantly associated with increased incidences of metastases at diagnosis and death (Figure 7A,B). The survival curve showed that high *ANRIL* expression was a significant predictor of a reduced overall survival rate (Figure 7C). *ANRIL* expression did not predict % necrosis (*p* = 0.2).

## 3. Discussion

The large variability observed in response to the standard of care treatment in osteosarcoma presents a strong need for clinically applicable biomarkers to determine a patient’s likelihood of response in order to allow for individualized treatment regimens upfront. To date, patient clinical characteristics, along with many protein-coding gene expression and mutation statuses, have been assessed for clinical utility, but none have been incorporated into clinical practice to predict chemotherapeutic responses [10]. An investigation into the non-protein-coding genome may be fruitful and present a new opportunity to predict chemosensitivity prior to the administration of several cycles of toxic and potentially ineffective chemotherapy.

As a new frontier in cancer biomarker study, very little is known about the biological and physio-pathological role of lncRNAs [39]. To that end, our group previously evaluated lncRNAs in relation to chemosensitivity utilizing available data from the Genomics of Drug Sensitivity in Cancer database (GDSC) and identified significant correlations between decreased cisplatin and doxorubicin sensitivity and increased lncRNA *ANRIL* expression. The lncRNA *ANRIL* is transcribed in the opposite direction from the IN-K4b–ARF–INK4a gene cluster [40,41]. The protein-coding genes located within this gene cluster include S-methyl-5′-thioadenosine phosphorylase (*MTAP*), *CDKN2A*, which encodes splice variants p16INK4A and p14ARF, and *CDKN2B*, which encodes p15INK4B. This gene locus is transcriptionally silenced or homozygously deleted at a frequency of nearly 40% in various tumors [42]. Previous studies have shown that *ANRIL* expression is upregulated in osteosarcoma tissues by comparison to adjacent normal tissues and may play a role in the regulation of cell proliferation, growth, and metastasis of osteosarcoma and other cancer types [30,31,32,43,44,45,46,47,48,49,50,51]. In relation to chemotherapeutic response, *ANRIL* has also been reported to promote cell chemoresistance, in multiple cancer types, to various therapeutic agents including cisplatin [33,34,35,36,37,52,53,54].

Based on our previous findings and existing evidence, we hypothesized that increased *ANRIL* expression may play a role in the chemoresistance of osteosarcoma to cisplatin and doxorubicin and that *ANRIL* knockdown may lead to increased chemosensitivity to those agents. Our in vitro experiments confirmed this hypothesis, showing increased sensitivity to both cisplatin and doxorubicin in siRNA *ANRIL*-mediated SAOS2 cells. Using *ANRIL* overexpression models, we further demonstrated that increased *ANRIL* expression resulted in a significantly decreased sensitivity to cisplatin and doxorubicin in U2OS, but not in the HOS cell line. Given the complex and diverse genomic background of the osteosarcoma cell lines employed for this study, it is possible that genes in addition to and/or other than *ANRIL* may impact the chemosensitivity in HOS cells. It is also possible that there is a gene-dose effect, in which a certain level of *ANRIL* expression needs to be reached for conversion to chemo-resistance. All of these warrant subsequent studies. In addition, our results also showed that increased *ANRIL* expression resulted in significantly increased rates of cell proliferation under drug-naïve conditions in U2OS and HOS cells. Though not statistically significant, SAOS2 *ANRIL* knockdown cells displayed a predicted trend of decreased proliferation rates under drug-naïve conditions. The effect of *ANRIL* expression on cancer cell proliferation observed in our study is consistent with previous reports documented in retinoblastoma, liver cancer, gastric cancer and lung cancer [30,31,32,33,34,35,36]. Additionally, a previous study by Li G. et al. showed decreased cell proliferation under drug-naïve conditions and increased cisplatin sensitivity in U2OS cells after *ANRIL* knockdown [34].

It is important to note that because of the close physical proximity of *ANRIL* to several nearby protein-coding genes (*CDKN2A*, *CDKN2B*, *MTAP*), and the existence of multiple transcriptional isoforms of *ANRIL*, the role of *ANRIL* on the potential regulation of these genes in relation to drug sensitivity has not been fully elucidated. Previous functional studies have shown that *ANRIL* can mediate the epigenetic transcriptional repression of the neighboring genes *CDKN2A* and *CDKN2B* [55,56]. However, multiple studies have also reported positive correlations between *ANRIL* and nearby protein-coding genes, suggesting transcriptional co-regulation of these genes predominates in many tissues [57,58,59,60,61]. We carefully designed our *ANRIL* modification experiments to only target *ANRIL* and showed that there were no significant gene expression changes in three neighboring protein-coding genes. Previously, one of these nearby protein-coding genes (*MTAP*) was identified in osteosarcoma [62]. It is thought to play a role in cancer cell etiology given its behavior as a tumor suppressor [62,63,64,65]. However, correlation analyses between the expression of *MTAP* and cisplatin and doxorubicin sensitivity found no relationship between them.

Our analysis of TARGET data confirmed an association between osteosarcoma prognosis and *ANRIL* expression. As predicted, median *ANRIL* expression was associated with death and metastases at diagnosis. This is consistent with previous reports indicating that high *ANRIL* expression is a poor prognosis indicator in different cancer types including osteosarcoma [44,66,67,68,69].

In this study, we identified and validated the link between *ANRIL* expression and cisplatin and doxorubicin sensitivity in osteosarcoma. However, the underlying mechanisms of this relationship remain unclear, which may require a more comprehensive, longitudinal examination of protein-coding genes, both at the RNA and protein level, after *ANRIL* modification. Indeed, previous studies have provided evidence for significant relationships with select microRNAs and regulated genes, such as let-7a/HMGA2, miRNA-125a-5p/STAT3, microRNA-98, and microRNA-328/ABCG2, to further elucidate potential mechanisms involved in ANRIL-mediated chemoresistance to cisplatin across different cancer types [31,32,33,34,35,36,52]. A separate study in oral squamous cell carcinoma also identified significant impairment of drug transporters MRP1 and ABCC2 after *ANRIL* knockdown, which resulted in increased cisplatin cytotoxicity [53].

Our study focused on *ANRIL*’s impact on cancer cell growth inhibition induced by chemotherapy. Additional cellular phenotypes can be examined to assess whether changing the *ANRIL* expression level will impact other traits (e.g., migration and invasion). Since *ANRIL* is unlikely to be the only lncRNA that contributes to chemosensitivity, subsequent animal studies are warranted to assess the effect of *ANRIL* in the osteosarcoma response to chemotherapy in vivo. Furthermore, the microRNAs that are negatively regulated by *ANRIL* (such as miR-34a, miR-125a, and miR-186) should be investigated, as should their putative role in bone remodeling and osteosarcoma onset [70]. Further studies would include the investigation of these targets in osteosarcoma models that are particularly related to chemosensitivity.

## 4. Materials and Methods

### 4.1. Reagents

Cisplatin and doxorubicin were purchased from Sigma-Aldrich (St. Louis, MO, USA) and dissolved in either cell culture-grade water at a stock concentration of 3 mM for cisplatin, or in DMSO at a stock concentration of 10mM for doxorubicin. Reconstituted stocks were aliquoted into 1.5 mL black centrifuge tubes to protect from light and stored at −80 °C as single-use aliquots prior to experimentation.

### 4.2. Cell Lines

The osteosarcoma cancer cell lines SAOS2, HOS, and U2OS were obtained from the American Type Culture Collection (ATCC, Rockefeller, MD, USA) and selected based on *ANRIL* expression profiling and somatic mutation status (Appendix A, Appendix A and Appendix A). All cell lines were grown using McCoy’s 5A modified medium (Thermo Fisher Scientific, Waltham, MA, USA), supplemented with 15% FBS (fetal bovine serum; Thermo Fisher Scientific) in a temperature-controlled incubator, and maintained at 37 °C with 5% CO_2_. Cells in the logarithmic phase of growth were used for all experiments. Cell lines were observed microscopically to confirm morphology, and population-doubling times were determined by viable cell counting using trypan blue on the TC20™ Automated Cell Counter (Bio-Rad Laboratories, Hercules, CA, USA). All cell lines were periodically monitored for mycoplasma using the Universal Mycoplasma Detection Kit following the manufacturer’s protocol (ATCC). Culture health and identity were monitored by microscopy and by comparing population doubling times to baseline values recorded at the time of receipt.

### 4.3. lncRNA ANRIL Knockdown in SAOS2 Cells

Knockdown of *ANRIL* was performed in the SAOS2 cell line, due to its high *ANRIL* expression, by utilizing Lincode SMARTpool small interfering RNA (siRNA) against *ANRIL* (*CDKN2B-AS1*) and the DharmaFECT transfection reagent 2 (Dharmacon, Horizon Discovery Biosciences, Boulder, CO, USA) following the manufacturer’s reverse transfection protocol. The Lincode Non-targeting Control Pool was used as a control in all siRNA knockdown experiments. Each siRNA pool consisted of four different siRNAs and was reconstituted to a 20 nM stock concentration. Knockdown efficiency was experimentally determined by reverse transfecting SAOS2 cells with 25 nM of either the *ANRIL* or control siRNA and 0.05% DharmaFECT transfection reagent 2 in a 6-well tissue culture plate, utilizing a seeding density of 5 × 10^5^ cells per well. Reverse transfection was performed in a total volume of 2 mL containing RPMI 1640 with 10% FBS and maintained at 37 °C in a humidified incubator with 5% CO_2_ atmosphere. Cells were harvested at 24, 48, and 72 h following reverse transfection to determine knockdown efficiency by assessing lncRNA *ANRIL* expression using quantitative real-time PCR.

### 4.4. lncRNA ANRIL Overexpression in HOS and U2OS Cells

Stable *ANRIL* overexpression models were created from the HOS and U2OS cell lines utilizing a custom designed Mammalian noncodingRNA Expression Lentiviral Vector (pLV(ncRNA)EGFP:T2A:Puro-CMV>hCDKN2B-AS1[NR_003529.3]; Vector ID:VB200211-1124awf) obtained from Vector Builder (VectorBuilder Inc., Chicago, IL, USA). A mCherry empty vector was utilized as a control (pLV[Exp]-EGFP:T2A:Puro-EF1A>mCherry; Vector ID: VB160109-10005). HOS and U2OS cells were plated at a seeding density of 1 × 10^5^ cells per well in a 6-well tissue culture plate and allowed to adhere for 24 h in a 37 °C humidified incubator with 5% CO_2_ atmosphere prior to transduction. Cells were transduced in 1 mL media containing the appropriate virus to obtain an MOI of 10 following the manufacturer’s protocol. Stable *ANRIL* expression lines and empty vector controls in both HOS and U2OS cells were generated following a 3-day antibiotic selection with puromycin. Positive GFP-expression cells were visualized and counted daily during selection to confirm transduction efficiency using the Cytation™ 1 Cell Imaging Multi-Mode Reader (BioTek Instruments, Winooski, VT, USA). Confirmation of target *ANRIL* expression was performed by assessing lncRNA *ANRIL* expression using quantitative real-time PCR.

### 4.5. lncRNA and Gene Expression Using Quantitative Real-Time PCR

Total RNA was extracted from cultured cells using the Quick-RNA MiniPrep Plus Kit (Zymo Research, Irvine, CA, USA) and quantified on the NanoDrop ND-8000 8-channel spectrophotometer (Thermo Fisher Scientific, Waltham, MA, USA). A total of 2 µg of RNA was used to synthesize cDNA, utilizing the High Capacity cDNA Reverse Transcription Kit (Thermo Fisher Scientific, Waltham, MA, USA). The Sso-Advanced Universal SYBR Green SuperMix (Bio-Rad, Hercules, CA, USA) was used to conduct real-time PCR analyses following the manufacturer’s protocol. In addition to the target lncRNA *ANRIL*, we assessed the expression of three protein-coding genes located near *ANRIL* to assess potential off-target effects following *ANRIL* knockdown or overexpression: *CDKN2A*, *CDKN2B*, and *MTAP*. *GAPDH* and *ACTB* (β-Actin) served as housekeeping genes. Pre-designed primers for all genes were selected from PrimeTime qPCR Primer Assays available from Integrated DNA Technologies, Inc. (IDT; Coralville, IA, USA). The PCR primer sequences utilized to assess gene expression are provided in Table 1.

Quantitative real-time PCR and data collection were performed using the 7500 Real-Time PCR System (Applied Biosystems, Foster City, CA, USA). All results were normalized with the expression of *GAPDH* and *ACTB* as a reference panel. All PCR reactions were performed in triplicate, and each knockdown and overexpression experiment was performed independently three times. Expression results were quantified using the ΔΔCt method relative to the appropriate control [71].

### 4.6. Cell Viability and Drug Response Curves in Knockdown and Overexpression Models following Cisplatin and Doxorubicin Exposure

To assess the impact of siRNA-mediated knockdown of the lncRNA *ANRIL* on proliferation and responses to cisplatin and doxorubicin, SAOS2 cells were reverse transfected in 96-well plates, with 25 nM of either the *ANRIL* or control siRNA and 0.05% DharmaFECT transfection reagent 2, using a seeding density of 1 × 10^4^ cells per well. Following a 24-h incubation in a 37 °C humidified incubator with 5% CO2 atmosphere, cells were treated with serial dilutions of either cisplatin at concentrations ranging from 1 µM to 14 µM, or of doxorubicin at concentrations ranging from 0.1 µM to 2 µM. Cell viability was measured at 24, 48, and 72 h after treatment by incubation with WST-1 (Roche Applied Science, Penzberg, Upper Bavaria, Germany) following the manufacturer’s protocol. Following a 2-h incubation at 37 °C after the addition of WST-1, absorbance at the 450 nm wavelength was assessed using the Synergy HTX Multi-Mode Plate Reader (BioTek, Winooski, VT, USA).

To assess the impact of *ANRIL* overexpression on proliferation and responses to cisplatin and doxorubicin, stable transduction models for the overexpression of *ANRIL*, or empty expression control generated in U2OS and HOS cell lines, were plated in 96-well plates using a seeding density of 1 × 10^4^ cells per well, and incubated for 24 h before treatment with serial dilutions of either cisplatin or doxorubicin. Cell viability was measured at 24, 48, and 72 h after treatment by incubation with WST-1 as described above.

Each experimental condition was performed in triplicate and measurements were reported as the mean ± standard deviation (S.D.) of three independent biological experiments.

### 4.7. ANRIL Expression and Clinical Outcomes Assessment

To investigate the impact of *ANRIL* expression on clinical outcome measures in osteosarcoma, we utilized available RNASeq data from the TARGET (Therapeutically Applicable Research to Generate Effective Treatments) osteosarcoma dataset (*n* = 86) (https://portal.gdc.cancer.gov/projects, dbGaP Study Accession: phs000468.v21.p8) accessed on 13 April 2021. This dataset is accessible via the National Cancer Institute Office of Cancer Genomics in the Genomic Data Commons (GDC). It includes tumors from children enrolled in Children’s Oncology Group (COG) protocols from which next-generation sequencing and transcriptome sequencing was performed to report long non-coding RNA expression. A summary of available clinical outcome measures and patient characteristics from TARGET osteosarcoma dataset are provided in Table 2.

### 4.8. Statistical Analysis

Statistical analyses of all in vitro cell culture experiments and patient data were performed in GraphPad Prism 6.07 (San Diego, CA, USA). For all cell culture experiments, significant differences between groups were tested using a two-tailed Student’s *t*-test or two-way analysis of variance (two-way ANOVA) [72]. For the patient cohort dataset, patients were categorized into high (>2) and low *ANRIL* expression groups based on the median *ANRIL* expression. Comparisons of ANRIL expression with clinical parameters were analyzed using the Student’s *t*-test or the Chi-square test [73]. Survival analysis was performed using the Kaplan–Meier method and the log-rank test [74,75]. A *p*-value of <0.05 was considered statistically significant.

## 5. Conclusions

The overexpression of *ANRIL* significantly increased cell proliferation and reduced both cisplatin and doxorubicin sensitivity, whereas the reduction in *ANRIL* expression led to increased sensitivity in our in vitro validation experiments. Analysis within the TARGET osteosarcoma patient cohort showed that higher *ANRIL* expression was significantly associated with death and metastases. These findings further support the hypothesis that *ANRIL* is a suitable chemo sensitivity and prognostic biomarker in osteosarcoma. Furthermore, reducing *ANRIL* expression may be a therapeutic strategy to overcome current standard-of-care treatment resistance in osteosarcoma.

## Figures and Tables

**Figure 1 ijms-22-11168-f001:**
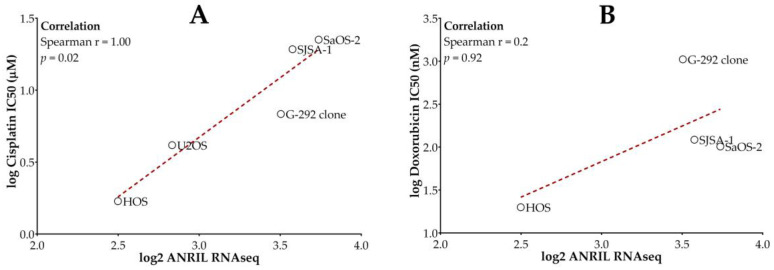
Increased *ANRIL* expression correlates with increased resistance to both cisplatin (**A**) and doxorubicin (**B**) in osteosarcoma cell lines. In vitro drug response data were obtained from the Genomics of Drug Sensitivity in Cancer version 1 dataset (GDSC1; https://www.cancerrxgene.org/ release 8.3, June 2020) accessed on 4 June 2020. Measured ANRIL expression was obtained from the Cancer Cell Line Encyclopedia (CCLE, https://portals.broadinstitute.org/ccle/data, release date 1 February 2019) accessed on 4 June 2020. X-axis displays *ANRIL* expression after log2 transformation. Y-Axis displays the reported therapeutic IC50 (µM) value after log transformation.

**Figure 2 ijms-22-11168-f002:**
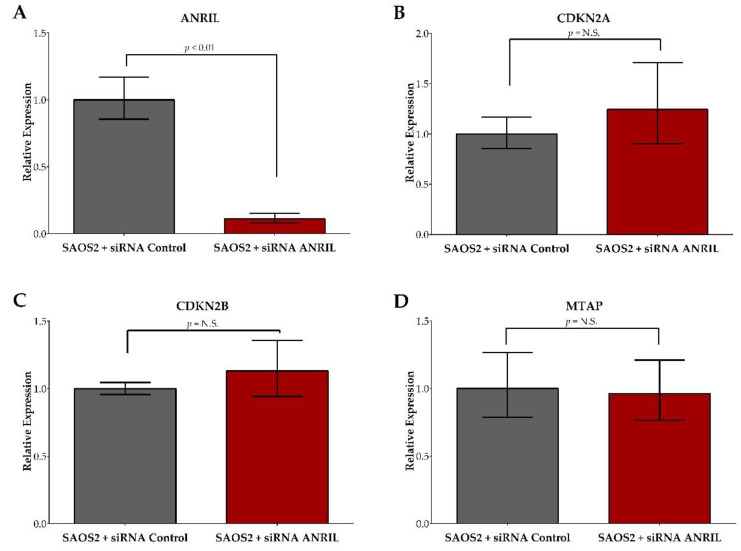
In vitro validation of lncRNA *ANRIL* knockdown in SAOS2 cells after siRNA transfection. SAOS2 cells transfected with *ANRIL* siRNA displayed an 88.9% decrease in *ANRIL* expression compared to control cells (**A**). No significant changes in gene expression of *CDKN2A* (**B**), *CDKN2B* (**C**), or *MTAP* (**D**) were observed between SAOS2 cells transfected with *ANRIL* siRNA and control. Red bars indicate relative expression generated from SAOS2 cells transfected with *ANRIL* siRNA. Grey bars indicate relative expression generated from SAOS2 transfected with control siRNA.

**Figure 3 ijms-22-11168-f003:**
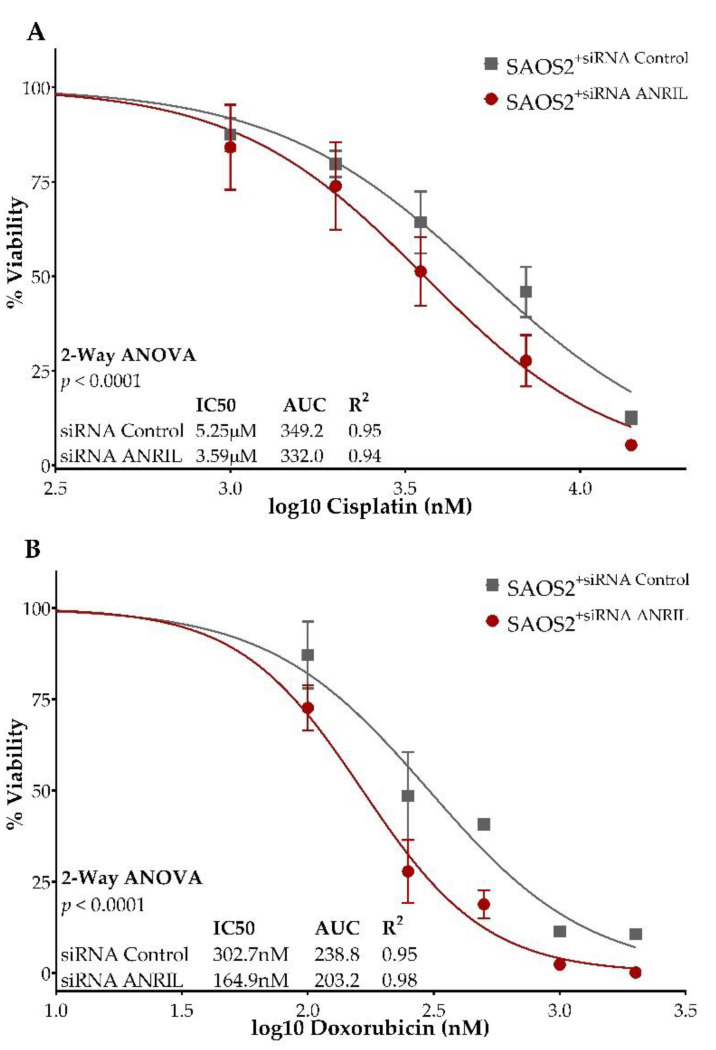
ANRIL knockdown increases the sensitivity to cisplatin and doxorubicin in osteosarcoma. Dose–response curves of SAOS2 cells transfected with either ANRIL siRNA or control measured at 72 h post-treatment with either cisplatin (**A**) or doxorubicin (**B**). Y-axis displays percent viability. X-axis displays log-transformed treatment concentrations. *ANRIL* knockdown SAOS2 cells are represented by lines and circles shaded in red. Control siRNA SAOS2 cells are represented by lines and squares shaded in grey. Each experimental condition was performed in triplicate and measurements were reported as the mean ± standard deviation (S.D.) of three independent biological experiments. Dose points with minimal error (S.D. < 2) show no visible error bars.

**Figure 4 ijms-22-11168-f004:**
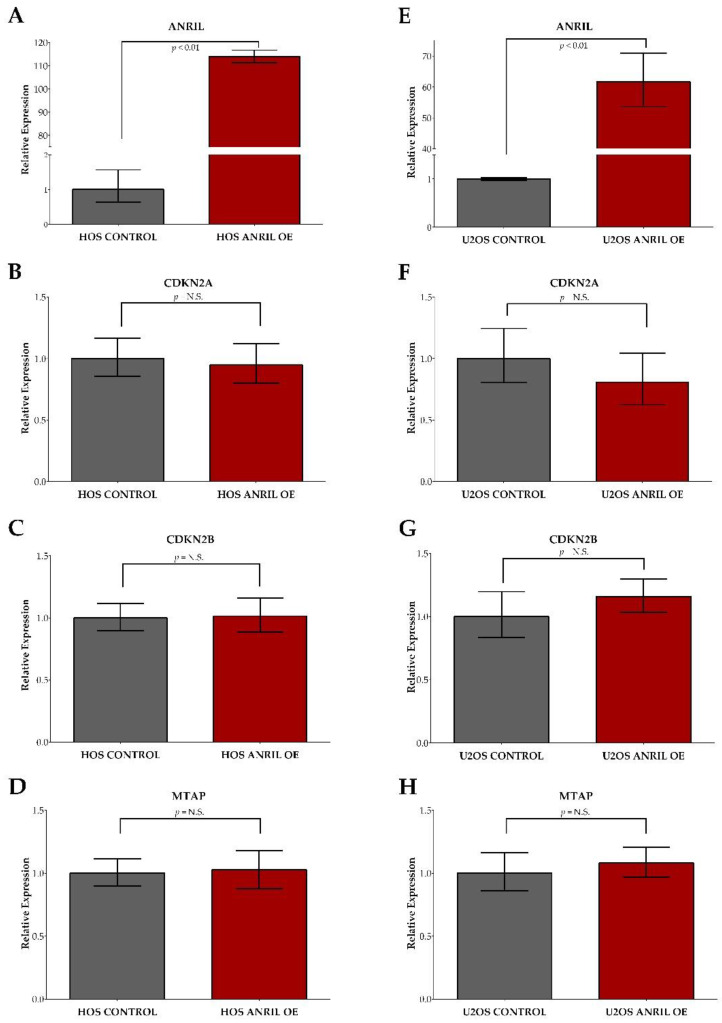
In vitro validation of lncRNA *ANRIL* overexpression in stable overexpression models. HOS cells containing stable *ANRIL* overexpression showed a 114-fold increase in ANRIL expression compared to the control (**A**). U2OS cells containing stable ANRIL overexpression showed a 60-fold increase in *ANRIL* expression compared to the control (**E**). No significant changes in gene expression of *CDKN2A*, *CDKN2B*, or *MTAP* were observed between the *ANRIL* overexpression and control models for HOS (**B**–**D**) or U2OS (**F**–**H**). Red bars indicate relative expression generated from *ANRIL* overexpression models. Grey bars indicate relative expression generated from empty vector controls.

**Figure 5 ijms-22-11168-f005:**
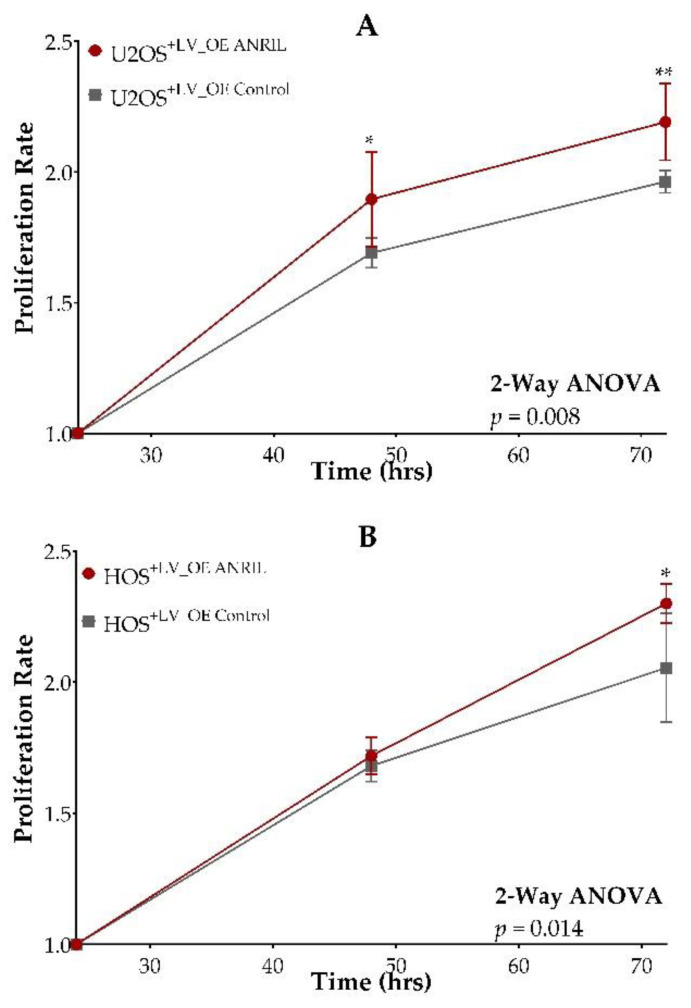
Overexpression of *ANRIL* increases cell proliferation. U2OS cell line models are shown in panel (**A**). HOS cell line models are shown in panel (**B**). Y-axis displays proliferation rate. X-axis displays time of measurement in hours. Calculated measurements of proliferation were normalized to measurements taken at 24 h after cell plating. *ANRIL* overexpression models are represented by lines and circles shaded in red. Empty vector expression control models are represented by lines and squares shaded in grey. * student *t*-test *p*-value < 0.05. ** student *t*-test *p*-value < 0.01.

**Figure 6 ijms-22-11168-f006:**
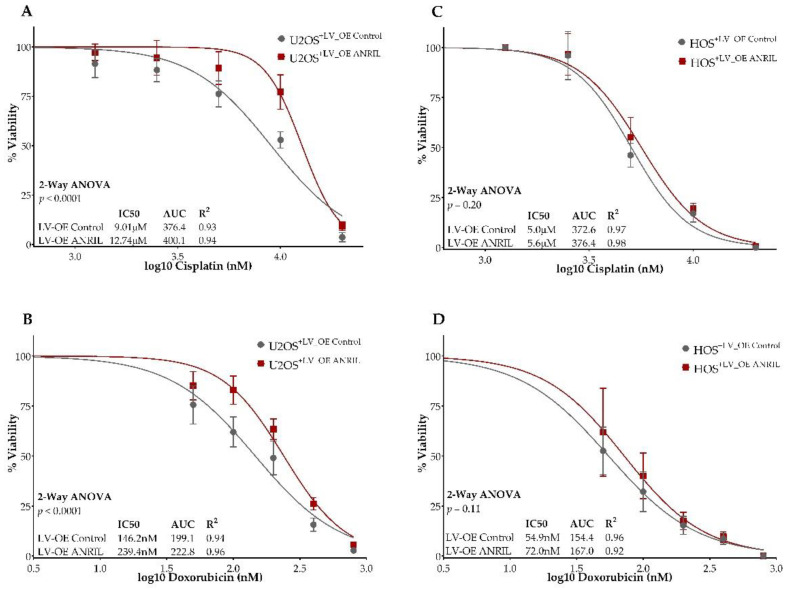
*ANRIL* overexpression decreases the sensitivity to cisplatin and doxorubicin in osteosarcoma. Dose–response curves of *ANRIL* overexpression U2OS lines treated with cisplatin or doxorubicin are displayed in panels (**A**,**B**). Dose–response curves of *ANRIL* overexpression HOS lines treated with cisplatin or doxorubicin are displayed in panels (**C**,**D**). Y-axis displays percent viability. X-axis displays log-transformed treatment concentrations. *ANRIL* overexpression models are represented by lines and circles shaded in red. Empty vector control models are represented by lines and squares shaded in grey. Each experimental condition was performed in triplicate and measurements were reported as the mean ± standard deviation (S.D.) of three independent biological experiments. Dose points with minimal error (S.D. < 2) show no visible error bars.

**Figure 7 ijms-22-11168-f007:**
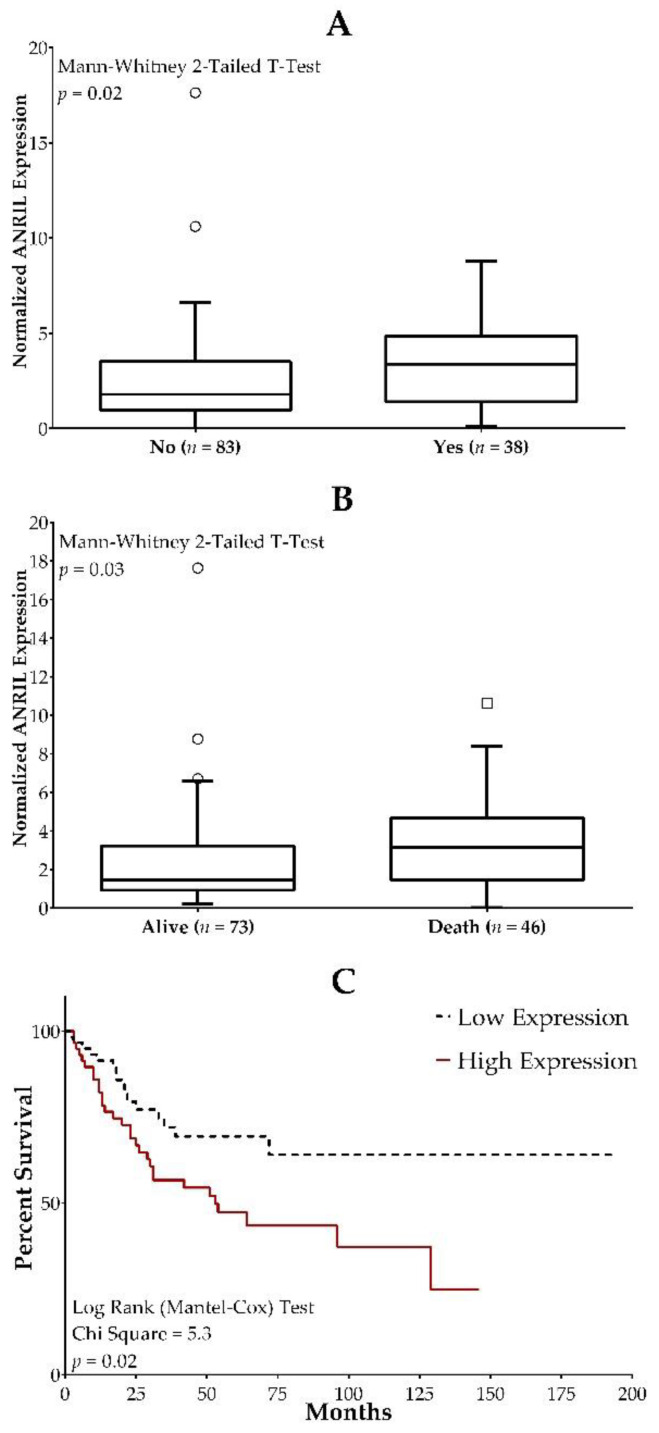
Associations between *ANRIL* expression and clinical outcomes in osteosarcoma. Overexpression of *ANRIL* is associated with increased incidences of metastases at diagnosis (**A**) and death (**B**). Survival curve analysis showed that high ANRIL expression was a significant predictor of a reduced overall survival rate (**C**).

**Table 1 ijms-22-11168-t001:** PCR primer sequences utilized to assess gene expression.

Gene	PCR Primer Sequence
*ANRIL*	F: 5′-CTCTCATCTGATCTCCGTCCT-3′
	R: 5′-TCACATCCAAGACAGCAAGT-3′
*CDKN2A*	F: 5′-GTGCCACATTCGCTAAGTG-3′
	R: 5′-ACCCTGTCCCTCAAATCCT-3′
*CDKN2B*	F: 5′-ATGCGTTCACTCCAATGTCT-3′
	R: 5′-CTCCACTTTGTCCTCAGTCTTC-3′
*MTAP*	F: 5′–GCAGCCATGCTACTTTAATGTC-3′
	R: 5′–GCTTACTGCTCACTACCATACC–3′
*GAPDH*	F: 5′-GAACATCATCCCTGCCTCTAC-3′
	R: 5′-CCTGCTTCACCACCTTCTT-3′
*ACTB* (β-Actin)	F: 5′-GTGGCCGAGGACTTTGATT-3′
	R: 5′-TTTAGGATGGCAAGGGACTTC-3′

F: forward. R: Reverse.

**Table 2 ijms-22-11168-t002:** Clinical characteristics and outcomes from the TARGET Osteosarcoma patient cohort.

TARGET Osteosarcoma Patient Dataset	
Clinical Characteristic	*n* (%) or Mean (Range)
**Sex**	** *n* ** **= 86**
Males	50 (58%)
Females	36 (42%)
**Age at Diagnosis, Mean (Range)**	**15 (8–32)**
<14 years of age	35 (41%)
≥14 years of age	51 (59%)
**Metastases at Diagnosis**	** *n* ** **= 86**
Yes	21 (24%)
No	65 (76%)
**Overall Survival in Months**	**126 (0–487)**
<36 months	73 (86%)
≥36 months	12 (14%)
**% Necrosis**	** *n* ** **= 43**
<91%	19 (44%)
≥91%	24 (56%)
**Death**	** *n* ** **= 86**
Yes	28 (33%)
No	58 (67%)

## Data Availability

The results published here are, wholly or in part, based upon data generated by the Therapeutically Applicable Research to Generate Effective Treatments (https://ocg.cancer.gov/programs/target, dbGaP Study Accession: phs000468.v21.p8) accessed on 13 April 2021. The data used for this analysis are available at https://portal.gdc.cancer.gov/projects accessed on 13 April 2021.

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
