# Peer review of "Long Non-Coding RNA ANRIL as a Potential Biomarker of Chemosensitivity and Clinical Outcomes in Osteosarcoma"

_ijms, 2021, doi:10.3390/ijms222011168_

Round 1

Reviewer 1 Report

The present manuscript hypothesizes a possible involvement of lncRNA ANRIL (anti-sense non-coding RNA in the INK4 locus) in the responsiveness to cisplatin and doxorubicin of osteosarcoma cells. The Authors aimed to propose ANRIL as a biomarker for chemotherapeutic sensitivity in osteosarcoma.

Complementary approaches were used to assess their hypotheses. The Authors used various in vitro models: human SaOS2, U2OS, HOS osteosarcoma cell lines modified for repression or overexpression of ANRIL, and bioinformatics analyses of publicly available datasets.

The manuscript is well written, methods are well detailed, results are convincing but the conclusions are not entirely in line with the results. Some additional experiments should also be conducted. In order for this paper to be fully suitable for publication, several points have to be considered.

The Authors evaluated the relationship between ANRIL expression levels and sensitivity to chemotherapy in cell lines models and in clinic dataset. Based on their modified cell models, a correlation could be suggested even if no positive correlation was detected for doxorubicin IC50 values in the osteosarcoma cell line panel (too few number of cell lines). However, the TARGET osteosarcoma database querying revealed that “ANRIL expression did not predict % necrosis (p=0.2)” [Line 373]. This histological response to preoperative chemotherapy is unfortunately an important clinical indicator to identify patients may be candidates for more intensive or novel therapy. The results thus do not support the hypothesis “that ANRIL may play a role in the variability of therapeutic response to cisplatin and doxorubicin and could serve as a potential clinical biomarker of therapeutic response in osteosarcoma” [Line 103-106]. They neither support the conclusion “that ANRIL is a suitable chemo sensitivity and prognostic biomarker in osteosarcoma”. [Line 467-468].

As mentioned line 52 and 61, methotrexate is another key chemodrug for osteosarcoma therapy regimen. The dose-dependent response of the different cell lines to MTX should be evaluated.

U2OS exhibiting an intermediate ANRIL expression level compared to SaOS2 and HOS, it should be interesting to complete the in vitro assays with ANRIL silencing.

Minor comment:

It is tricky to mix the terms “cell growth”, “cell proliferation” and “cell viability” [Line 269-271] as they refer to different cell process. The manufacturers present WST1 as a Cell Proliferation Reagent, but the assay evaluates the catalytic activity of some mitochondrial enzymes, which reflects the cell metabolic activity rather than cell proliferation (related to doubling or DNA replication).

Reviewer 2 Report

The manuscript titled “Long non-coding RNA ANRIL as a potential biomarker of chemosensitivity and clinical outcomes in osteosarcoma” by Dr. Ferdjallah and colleagues investigated the functional role of the Long non-coding RNA ANRIL in osteosarcoma as well as its role as prognostic marker. Functional results obtained in vitro indicated that the inhibition of ANRIL overexpression in osteosarcoma cell lines SAOS2 can induce a significant increase in cellular sensitivity to both cisplatin and doxorubicin, while overexpression of ANRIL in both HOS and U2OS cells can led to an increased resistance to both agents. As a biomarker, high levels of ANRIL expression were significantly associated with increased rate of metastases at diagnosis and death and was a significant predictor of reduced overall survival rate. In my opinion, the study is in general well organized and well presented. Results are interesting. The overall quality of the study and quality of the writing is adequate for publication. I have several comments

Major points
1.    Methods are completely lacking in supporting references. Please include references.
2.    The negatively regulated miRNAs by ANRIL (for instance miR-34a, miR-125a and miR-186 PMID: 33528317) as well as their putative role in bone remodeling and/or osteosarcoma onset, if present, should be included in the discussion section. Investigating these targets in osteosarcoma study models, including cisplatin chemoresistance models, can be part of further studies

Minor comments
Line 50 “methotrexate.[3].” Please remove the additional period
Linr 63 better prognostic biomarkers?
Lines 72-82 LncRNAs play a key role in osteogenic differentiation (PMID: 33898434). Consistently, a growing number of LncRNAs has been found to be linked to bone tumor onset/development (PMID: 34439367; PMID: 33433358). These notions and supporting references should be included
Lines 111-112 Please include a space between lines
Lines 175-183 May I suggest including the primers’ sequences in a table? Have these sequences been obtained from previous studies? In this case, supporting references should be uncluded
Lines 367-368 “Survival outcomes, metastases at diagnosis, and percent necrosis were evaluated in osteosarcoma patients (n=121 and n=43 for OS/metastases and percent necrosis, respectively)” Is this the TARGET cohort whose characteristics are reported in table 1? Table one depicts the characteristics of n=86 patients, only. Characteristics of the remaining patients should be included
Line 390 better lncRNAs?

Round 2

Reviewer 1 Report

NA